# Dual-Hybrid Attention Network for Specular Highlight Removal

Xiaojiao Guo*†
University of Macau
Macau, China
yc27441@um.edu.mo

Xuhang Chen*‡§
Huizhou University
Huizhou, China
xuhangc@hzu.edu.cn

Shenghong Luo
University of Macau
Macau, China
ykd823332077@gmail.com

Shuqiang Wang¶
Shenzhen Institute of Advanced
Technology, Chinese Academy of
Sciences
Shenzhen, China
sq.wang@siat.ac.cn

Chi-Man Pun¶
University of Macau
Macau, China
cmpun@umac.mo

## Abstract

Specular highlight removal plays a pivotal role in multimedia applications, as it enhances the quality and interpretability of images and videos, ultimately improving the performance of downstream tasks such as content-based retrieval, object recognition, and scene understanding. Despite significant advances in deep learning-based methods, current state-of-the-art approaches often rely on additional priors or supervision, limiting their practicality and generalization capability. In this paper, we propose the Dual-Hybrid Attention Network for Specular Highlight Removal (DHAN-SHR), an end-to-end network that introduces novel hybrid attention mechanisms to effectively capture and process information across different scales and domains without relying on additional priors or supervision. DHAN-SHR consists of two key components: the Adaptive Local Hybrid-Domain Dual Attention Transformer (L-HD-DAT) and the Adaptive Global Dual Attention Transformer (G-DAT). The L-HD-DAT captures local inter-channel and inter-pixel dependencies while incorporating spectral domain features, enabling the network to effectively model the complex interactions between specular highlights and the underlying surface properties. The G-DAT models global inter-channel relationships and long-distance pixel dependencies, allowing the network to propagate contextual information across the entire image and generate more coherent and consistent highlight-free results. To evaluate the performance of DHAN-SHR and facilitate future research in this area, we compile a large-scale benchmark dataset comprising a diverse range of images with varying levels of specular highlights.

*Both authors contributed equally to this research.
†Also with Baoshan University.
‡Also with University of Macau.
§Also with Shenzhen Institute of Advanced Technology, Chinese Academy of Sciences.
¶Corresponding authors.

Through extensive experiments, we demonstrate that DHAN-SHR outperforms 18 state-of-the-art methods both quantitatively and qualitatively, setting a new standard for specular highlight removal in multimedia applications. The code and dataset are available at https://github.com/CXH-Research/DHAN-SHR.

## CCS Concepts

• **Computing methodologies → Reconstruction**; **Computational photography**.

## Keywords

Specular Highlight Removal, Dual-Hybrid Attention, Spatial and Spectral

**ACM Reference Format:**
Xiaojiao Guo, Xuhang Chen, Shenghong Luo, Shuqiang Wang, and Chi-Man Pun. 2024. Dual-Hybrid Attention Network for Specular Highlight Removal. In *Proceedings of the 32nd ACM International Conference on Multimedia (MM '24), October 28-November 1, 2024, Melbourne, VIC, Australia.* ACM, New York, NY, USA, 9 pages. https://doi.org/10.1145/3664647.3680745

## 1 Introduction

Specular highlights, the intense reflections of light sources on shiny surfaces, pose significant challenges in multimedia and computer vision applications. These reflections disrupt visual consistency, obscuring details and altering color fidelity, which can impact applications like video editing, content-based retrieval, and interactive media. Removing specular highlights is crucial for accurate image and video processing, yet it remains complex due to variable light conditions, surface properties, and angles of observation.

Traditional highlight removal techniques, based on models such as the dichromatic reflection model [1], often fall short in diverse real-world scenarios. Deep learning approaches have shown promise but typically require prior information, such as highlight masks, limiting their practicality. Current methods also struggle to effectively restore highlighted areas, leading to suboptimal results.

We propose an end-to-end network that eliminates the need for additional priors, enabling specular highlight removal in a single step while preserving the visual fidelity of the restored image. Our approach leverages the observation that only a small portion of the surface produces highlights due to its smoothness and reflection angle, while most areas remain diffuse. We aim to learn a global

relationship on illumination and color and consider local features to restore details such as texture in highlight regions. Our network design incorporates two dimensions of self-attention to capture both global and local-level relationships.

To capture local feature relationships, we introduce the Adaptive Local Hybrid-Domain Dual Attention Transformer. This sets our approach apart from current methods that focus solely on the spatial domain. Our transformer leverages frequency domain features to aid the learning process, uncovering fine details and overarching patterns. A window-based dual attention mechanism focuses on local inter-channel and inter-pixel relationships, significantly reducing computational complexity compared to traditional self-attention mechanisms. This allows the transformer to effectively capture fine-grained details and textures.

To further enhance the transformer's ability to capture dependencies across window boundaries, we incorporate a window-shifting mechanism which is inspired by Swin Transformer [2] and SwinIR [3]. By shifting the pixels to create new window partitions, our module facilitates information exchange between adjacent windows. This shifting operation allows the transformer to capture local dependencies that span across window boundaries, ensuring a seamless integration of features across the entire image.

For global-level dependencies, we propose the Channel-Wise Contextual Attention Module that employs efficient Transformers to capture inter-channel relationships, rather than inter-patch relationships. This design choice serves two purposes: firstly, it allows the network to focus on more coarse-grained whole features, and secondly, it avoids the high computational and memory demands associated with the original Vision Transformer [4]. By attending to the channel-wise context, our module can effectively capture the global dependencies between different feature maps, enabling the network to reason about the overall illumination and color distribution.

To better organize feature learning at different scales, we adopt a network architecture similar to UNet, strategically placing modules that focus on different granularities at various positions within the network. Modules that capture detailed information are placed at higher levels, while modules that focus on global information are positioned at lower levels. This hierarchical arrangement enables our network to effectively learn and process features at different scales, improving its ability to handle complex specular highlight removal tasks.

To provide a standardized basis for comparison and yield meaningful insights into the performance improvements of specular highlight removal methods, we assembled an extensive dataset by combining images from three different highlight removal datasets (PSD [5], SHIQ [6] and SSHR [7]). PSD features high-quality real-world ground truth images, while the other two datasets include generated reference ground truths and fully synthetic data, broadening the diversity of the training samples. We retrained seven state-of-the-art deep learning specular highlight removal methods on this unified benchmark and evaluated their performance, along with 11 traditional methods, on the test sets of the benchmark. The experimental results demonstrate that our approach outperforms 18 other state-of-the-art methods across various test datasets and metrics, showcasing its superiority in specular highlight removal.

Overall, our contributions can be summarized as follows:

- We propose the Dual-Hybrid Attention Network for Specular Highlight Removal (DHAN-SHR), an end-to-end specular highlight removal network that introduces novel hybrid attention mechanisms, including the Adaptive Local Hybrid-Domain Dual Attention and the Adaptive Global Dual Attention. These attention mechanisms enable DHAN-SHR to effectively and efficiently capture both spatial and spectral information, as well as contextual relationships at different scales, accurately removing specular highlights while restoring underlying diffuse components.
- We compile a comprehensive benchmark dataset for specular highlight removal by combining images from three different datasets, resulting in 29,306 training pairs and 2,947 testing pairs. We retrain and test 18 state-of-the-art methods on this new benchmark, conducting a thorough comparative analysis and laying a solid foundation for future advancements in the field.
- Demonstrating through extensive experiments that DHAN-SHR outperforms state-of-the-art methods, setting a new standard in image enhancement and specular highlight removal.

## 2 Related Work

Specular highlight removal is a subset of image restoration tasks that have evolved from traditional models to learning-based approaches. Recent innovations in related image restoration tasks include the use of frequency transforms [8–14], generative AI [15–19], lightweight models [20–22], and memory augmentation techniques [23]. Additionally, advancements have been made through adaptations from large models [24]. The process of specular highlight removal follows a similar trajectory.

### 2.1 Traditional Approaches

Traditional methods typically rely on physical models, color, or texture information to capture the relationship between diffuse reflection and specular reflection. These methods utilize optical principles and geometric relationships to derive highlight characteristics and perform detection and removal. Early work included illumination-based constraints [25] and color recognition [26]. Chromaticity analysis methods were advanced by Shen *et al.* [27, 28], while Yang *et al.* [29] and Shen *et al.* [30] leveraged diffuse reflection characteristics. The dichromatic reflection model [1] was used by Akashi *et al.* [31] and Souza *et al.* [32].

Additional methods include semi-automatic algorithms by Nurutdinova *et al.* [33], the L0 criterion approach by Fu *et al.* [34], high-pass filter techniques by Yamamoto *et al.* [35], and combined methods for HDR image generation by Saha *et al.* [36]. Wen *et al.* [37] used polarization information for iterative optimization in separation strategies.

Overall, traditional methods have evolved significantly, incorporating chromaticity analysis, diffuse reflection characteristics, and polarization to improve highlight removal.

### 2.2 Deep Learning-based Approaches

In recent years, learning-based methods have gained significant attention due to their ability to learn complex relationships between

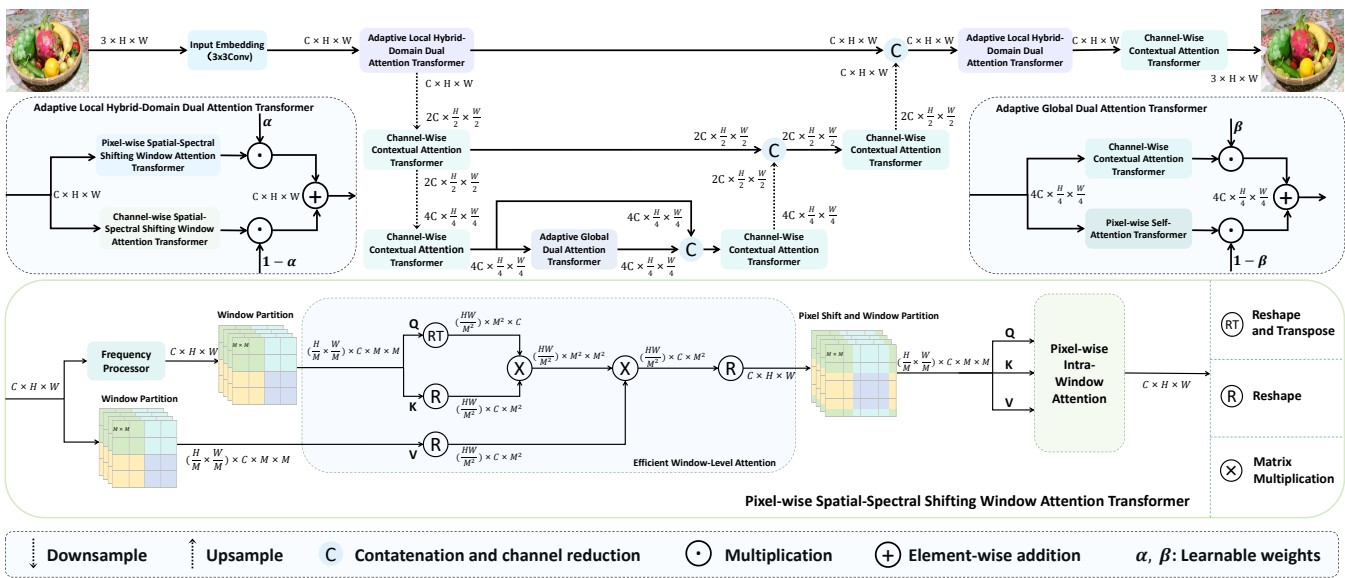

**Figure 1: The overall architecture of our proposed Dual-Hybrid Attention Network for Specular Highlight Removal (DHAN-SHR).**

input images and desired outputs. These methods often employ techniques such as Generative Adversarial Networks (GANs) [38] and Vision Transformer (ViT) [4] to address highlight removal.

Guo *et al.* [39] introduced SLRR, which laid the groundwork for subsequent deep learning-based methods, while Hou *et al.* [40] combined detection and removal networks. Wu *et al.* [5] developed SpecularityNet using GANs and attention mechanisms, and Fu *et al.* [6] proposed a multi-task network for enhanced performance. Liang *et al.* [41] combined mirror separation and intrinsic decomposition using adversarial networks.

Several other deep learning models have been developed to address highlight removal, including Unet-Transformer [42], which employs a highlight detection module as a mask to guide the removal task, and MG-CycleGAN [43], which leverages a mask generated by removing specular highlights from unpaired data to guide Cycle-GAN in transforming the problem into an image-to-image translation task. TSHRNet [7] has demonstrated superior performance in scenarios involving multiple objects and complex lighting conditions, while SHMGAN [44] is a neural network framework capable of effectively separating specular highlight maps and mirror distribution maps without the need for manual input labels.

Despite the advancements made by these state-of-the-art methods, they often encounter issues such as color inconsistency between highlight regions and the background, as well as the generation of unrealistic content within the highlight regions. To address these challenges, Hu *et al.* [45] proposed a neural network framework that effectively mitigates these problems, further pushing the boundaries of highlight removal techniques.

In summary, the field of highlight removal has witnessed significant progress through the development of deep learning methods. However, there remains room for improvement in terms of color consistency and the generation of realistic content within highlight regions, which future research should aim to address.

## 3 Methodology

### 3.1 Overall Architecture

Figure 1 illustrates the architecture of our proposed "DHAN-SHR", an end-to-end, one-stage network that takes a single image with specular highlight as input, without requiring any additional information such as highlight masks or priors. The network adopts a U-shape encoder-bottleneck-decoder structure to apply scale-specific feature learning methods at different levels.

The network begins by applying Adaptive Local Hybrid-Domain Dual Attention Transformers (L-HD-DAT) to low-level, high-resolution feature maps, capturing spatial and spectral information with two levels of local attention. The encoder pathway progressively downsamples the image, using Channel-Wise Contextual Attention Transformers (CCAT) to capture contextual information at lower resolutions. At the bottleneck, Adaptive Global Dual Attention Transformers (G-DAT) capture high-level semantic features. In the decoder pathway, the encoded features are gradually upsampled, recovering spatial details and fine-grained information. To maintain a balance between the encoder and decoder, we employ the same modules used in the corresponding levels of the encoder. And By concatenating these features via skip connections, the network can effectively combine the strengths of both pathways.

The network's architecture, with its scale-specific feature learning methods, enables effective capture and processing of features at different scales and semantic levels, leading to accurate and efficient specular highlight removal.

### 3.2 Adaptive Local Hybrid-Domain Dual Attention Transformer (L-HD-DAT)

The Adaptive Local Hybrid-Domain Dual Attention Transformer (L-HD-DAT) is a crucial component of our proposed DHAN-SHR

network, situated at the topmost position of the U-shaped architecture. Its primary objective is to process feature maps with the highest resolution, matching that of the input image, which contains the most abundant details. In the context of specular highlight removal, accurately detecting and removing specular highlights while restoring the corresponding diffuse visuals with consistent texture and detailed colors is critical for achieving high-quality results. This task is often considered the last mile in determining the visual quality of the highlight removal process.

The L-HD-DAT employs two parallel attention mechanisms to capture both inter-channel and inter-pixel relationships within local windows. These attention mechanisms are implemented through the Pixel-wise Spatial-Spectral Shifting Window Attention Transformer (P_SSSWAT) and the Channel-wise Spatial-Spectral Shifting Window Attention Transformer (C_SSSWAT). To adaptively adjust the contribution of each attention mechanism during the training process, we introduce a learnable weight coefficient $\alpha$. The L-HD-DAT can be formulated as follows:

$$\text{L-HD-DAT}(\mathbf{F}) = \alpha \times \text{P\_SSSWAT}(\mathbf{F}) + (1 - \alpha) \times \text{C\_SSSWAT}(\mathbf{F}), \quad (1)$$

where $\mathbf{F}$ represents the input features.

Both P_SSSWAT and C_SSSWAT follow the same procedure, which can be described as:

$$\mathbf{Y} = \mathbf{F} + \text{SSSWA}(\text{LN}(\mathbf{F}), \text{LN}(\text{FP}(\mathbf{F}))),$$
$$\text{SSSWAT}(\mathbf{F}) = \mathbf{Y} + \text{FFN}(\text{LN}(\mathbf{Y})). \quad (2)$$

In this procedure, $\mathbf{F}$ denotes the input features with dimension $C \times H \times W$, LN represents the LayerNorm operation, and FP is the Frequency Processor. The Spatial-Spectral Shifting Window Attention (SSSWA) is a key component of both P_SSSWAT and C_SSSWAT. The Feed Forward Network (FFN) consists of three convolutional layers that further process the attended features, enabling the network to capture complex spatial relationships and refine the feature representations.

*3.2.1 P_SSSWA: Pixel-wise Spatial-Spectral Shifting Window Attention.* Both P_SSSWAT and C_SSSWAT calculate intra-window attentions twice sequentially: first on the original features and then on the shifted features. The Pixel-wise Spatial-Spectral Shifting Window Attention (P_SSSWA) calculation procedure is illustrated in the lower half of Figure 1, with the first cascaded attention depicted in detail. The second attention calculation follows the similar approach, differing only in the input feature maps and the apply of an attention mask. Given input features $\mathbf{F} \in \mathbb{R}^{C \times H \times W}$, we first obtain the corresponding spectral features $\mathbf{F_s}$ with the same dimension $C \times H \times W$ using the Frequency Processor. Both $\mathbf{F_s}$ and $\mathbf{F}$ are then partitioned into non-overlapping $M \times M$ windows, similar to SwinIR [3], where $M$ represents the window height or width (in pixels). To ensure consistent window sizes, we pad the feature maps' right and bottom edges with zeros before partitioning. After partitioning and reshaping, $\mathbf{F_s}$ and $\mathbf{F}$ have dimensions $\frac{HW}{M^2} \times C \times M^2$, where $\frac{HW}{M^2}$ represents the total number of windows in a single channel.

Next, we project the spectral features $\mathbf{F_s}$ to $\mathbf{Q_s}$ (query) and $\mathbf{K_s}$ (key) and compute their self-attention, enabling the model to capture fine details and subtle variations that may be challenging to discern in the spatial domain alone. The resulting spectral attention, with dimensions $\frac{HW}{M^2} \times M^2 \times M^2$, is then multiplied with the spatial features $\mathbf{F}$ (as $\mathbf{V}$ (value)), allowing the model to selectively

attend to relevant spatial regions based on insights gained from the frequency domain analysis. In short, the first cascaded Pixel-wise Spatial-Spectral Window Attention is computed as:

$$\text{P\_SSSWA\_1}(\mathbf{Q_s}, \mathbf{K_s}, \mathbf{V}) = \mathbf{V} \cdot \text{Softmax}\left(\frac{\mathbf{Q_s}^T \cdot \mathbf{K_s}}{M}\right). \quad (3)$$

By integrating information from both the spatial and spectral domains, this attention mechanism enables the model to effectively identify and remove specular highlights while preserving the underlying surface details, resulting in improved specular highlight removal performance.

Before the partition-reversed output features of P_SSSWA_1 are cascadedly input to the second attention block of P_SSSWAT, we perform a cyclic shift on the pixels of each feature map by $s$ pixels in both horizontal and vertical directions, to cover the areas on the boundaries of the windows from the first partition. This shifting rule is illustrated using four color blocks in Figure 2. Unlike Swin Transformer's [2] patch-based window partition rules, we directly partition the windows on the pixels themselves. This pixel-level window partitioning allows the window-based attention mechanism to reflect more detailed textures, as it operates on the raw pixel values rather than on abstracted patches.

We then partition and reshape the shifted features $\mathbf{F_{sh}} \in \mathbb{R}^{C \times H \times W}$ into dimensions $\frac{HW}{M^2} \times C \times M^2$, similar to the previous attention block, and project it to obtain $\mathbf{Q_{sh}}$, $\mathbf{K_{sh}}$ and $\mathbf{V_{sh}}$ with the same dimension $\frac{HW}{M^2} \times C \times M^2$. The self-attention of the shifted window is calculated by:

$$\text{P\_SSSWA\_2}(\mathbf{Q_{sh}}, \mathbf{K_{sh}}, \mathbf{V_{sh}}) = \mathbf{V_{sh}} \cdot \text{Softmax}\left(\frac{\mathbf{Q_{sh}}^T \cdot \mathbf{K_{sh}}}{M} + \mathbf{Mask}\right). \quad (4)$$

In addition to the different input features, the second cascaded attention differs in the inclusion of an attention mask used to distinguish pixels that were originally not adjacent but are now in the same window due to the cyclic shifting. This can be observed on the right and bottom sides of the shifted features in Figure 2. The mask has the same size as the input shifted feature maps $\mathbf{F_{sh}}$, ensuring a one-to-one position correspondence. We assign a value of 0 to the positions in the windows where the pixels were originally adjacent, and a value of $-100$ to the positions in the windows where the pixels were originally not adjacent. Figure 2 provides an intuitive understanding of these mask values. By adding this mask to the self-attention of $\mathbf{Q_{sh}}$ and $\mathbf{K_{sh}}$, the model maintains the relationship between originally adjacent pixels while suppressing the influence of pixels that are only adjacent due to cyclic shifting.

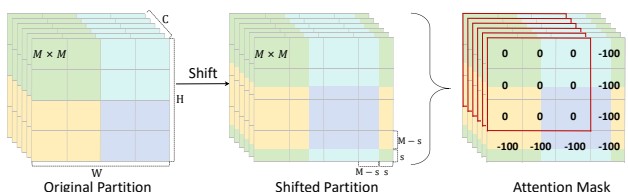

**Figure 2: Illustration of the window shifting approach and the attention mask applied to the pixel-wise shifting window attention.**

Since the attention computation is limited to non-overlapping windows, it significantly reduces the computational complexity

compared to traditional self-attention mechanisms that consider the entire image. In traditional approaches, the computational complexity is usually quadratic relative to the image size, and the computational cost is especially tremendous for high-resolution feature maps. However, for one calculation of intra-window pixel-wise attention with input features of dimension $C \times H \times W$ and a window size of $M \times M$, the approximate number of operations, such as element-wise multiplication and addition, is $(HW)M^2(2C-1) + (HW)C(2M^2-1)$. Since both M and C are constants, it means it has a linear computational complexity with respect to the image size. So the intra-window pixel attention not only helps to concentrate on local fine details but also offers computational efficiency.

*3.2.2 C_SSSWA: Channel-wise Spatial-Spectral Shifting Window Attention.* The working process of the C_SSSWA is similar to the previously described P_SSSWA. However, the main difference lies in the inter-channel window range attention calculation approach, which differs from P_SSSWA's inter-pixel counterpart. The input features remain the same, with $\mathbf{F} \in \mathbb{R}^{C \times H \times W}$ representing the spatial domain features and $\mathbf{F_s} \in \mathbb{R}^{C \times H \times W}$ representing the corresponding spectral domain feature maps.

Following the same procedure as P_SSSWA, we perform window partitioning, reshaping, and projection of $\mathbf{F}$ and $\mathbf{F_s}$ to obtain the corresponding window-based features $\mathbf{Q_s}, \mathbf{K_s}, \mathbf{V}$ in the same dimension $\frac{HW}{M^2} \times C \times M^2$. The first attention block, which hybridizes the spectral and spatial domain features, is calculated as:

$$C\_SSSWA\_1(\mathbf{Q_s}, \mathbf{K_s}, \mathbf{V}) = \text{Softmax}\left(\frac{\mathbf{Q_s} \cdot \mathbf{K_s}^T}{\tau}\right) \cdot \mathbf{V}, \quad (5)$$

where $\tau$ acts as a learnable temperature parameter that modulates the magnitude of the dot product. The attention result of $\mathbf{K_s}$ to $\mathbf{Q_s}$ has a dimension of $\frac{HW}{M^2} \times C \times C$, representing the interrelationships between the C channels for each window among the total $\frac{HW}{M^2}$ windows.

In the next step, we reverse the result of equation (5) with window partitioned dimension $\frac{HW}{M^2} \times C \times M^2$ back to the input feature maps' dimension $C \times H \times W$. After shifting the feature maps by s pixels, we partition, reshape, and project the new window-based features as $\mathbf{Q_{sh}}, \mathbf{K_{sh}}$, and $\mathbf{V_{sh}}$ for the second attention block, which is similar to P_SSSWA in employing shifted windows to capture dependencies across window boundaries, as shown in equation (6). However, unlike P_SSSWA, we do not use an attention mask as in equation (4). This is because when calculating the inter-channel relationships, we multiply the pixels that are in aligned positions within the window and then add them together. Therefore, the computation is not dependent on the pixels' positional relationships.

$$C\_SSSWA\_2(\mathbf{Q_{sh}}, \mathbf{K_{sh}}, \mathbf{V_{sh}}) = \text{Softmax}\left(\frac{\mathbf{Q_{sh}} \cdot \mathbf{K_{sh}}^T}{\tau}\right) \cdot \mathbf{V_{sh}}, \quad (6)$$

where $\mathbf{Q_{sh}}, \mathbf{K_{sh}}, \mathbf{V_{sh}} \in \frac{HW}{M^2} \times C \times M^2$, and $\tau$ is still a learnable temperature parameter. The computational complexity of both equation (5) and equation (6) is approximately $4C^2(HW)$, which is linearly related to the size of the image $H \times W$.

*3.2.3 Frequency Processor.* The Frequency Processor employed in both P_SSSWAT and C_SSSWAT generates feature maps that have undergone transforms to the frequency domain, as shown in Algorithm 1. First, we compute the discrete Fourier transform of the input feature maps. Then, a shallow convolution and GeLU

---

**Algorithm 1** Frequency Processor

**Require:** F (input features)
**Ensure:** $\mathbf{F_s}$ (frequency processed features)
1: Apply convolution: $\mathbf{identity_1} \leftarrow Conv2d_{1\times1}(\mathbf{F})$
2: Apply convolution: $\mathbf{identity_2} \leftarrow Conv2d_{1\times1}(\mathbf{F})$
3: Compute FFT of **F** and keep the real part: $\mathbf{F_{fft}} \leftarrow \text{FFT}(\mathbf{F}, \text{dim} = (-2, -1)).\text{real}$
4: Apply convolution to $\mathbf{F_{fft}}$: $\mathbf{F_{fft}} \leftarrow \text{GELU}(Conv2d_{1\times1}(\mathbf{F_{fft}}))$
5: Pass through MLP layers: $\mathbf{F_{fft}} \leftarrow \text{MLPs}(\mathbf{F_{fft}})$
6: Compute inverse FFT: $\mathbf{F_{ifft}} \leftarrow \text{IFFT}(\mathbf{F_{fft}}, \text{dim} = (-2, -1)).\text{real}$
7: Add residual connection: $\mathbf{F_s} \leftarrow \mathbf{F_{ifft}} + \mathbf{identity_2}$
8: Apply toning: $\mathbf{F_s} \leftarrow \text{Toning}(\text{Concat}([\mathbf{F_s}, \mathbf{identity_1}], \text{dim} = 1))$

---

activation are applied to the frequency features to introduce non-linearity and enhance the expressiveness of the frequency-domain representations. Before executing the inverse Fourier Transform, MLP layers are applied to the frequency-domain features to adapt and refine the spectral representations, capturing more accurate and contextually relevant frequency-domain information essential for reconstructing the diffuse visuals.

To provide conditional guidance during the spectral features' training process, we obtain two identities of the input spatial features, which are then added and concatenated, respectively, to the inverse Fourier Transform of the spectral features. Finally, a toning operation is applied to the spectral features, which removes the phase information and focuses on the magnitude-based spectral features, helping to suppress noise and outliers while emphasizing relevant frequency ranges for accurate and coherent diffuse visual reconstruction.

## 3.3 Channel-Wise Contextual Attention Transformer (CCAT)

The Channel-Wise Contextual Attention Transformers (CCAT) scale up its concentrated grain compared to L-HD-DAT by their strategic deployment within both the encoder pathway, which progressively downsamples, and the decoder pathway, which conversely upsamples. This positioning allows CCAT to operate on feature maps that possess a higher semantic level than the original input image. To effectively address the challenge of contextual information extraction across channels, CCAT incorporates an attention mechanism that is meticulously designed to weigh the significance of each channel based on its contextual relationship with others. This is achieved through a multi-headed self-attention module as equation (7) that operates on the channel dimension, enabling the network to dynamically adjust the emphasis on different channels according to their contextual relevance.

$$\text{CCA}(\hat{\mathbf{Q}}, \hat{\mathbf{K}}, \hat{\mathbf{V}}) = \text{Softmax}\left(\frac{\hat{\mathbf{Q}} \cdot \hat{\mathbf{K}}^T}{\sqrt{\hat{C}/h}}\right) \cdot \hat{\mathbf{V}}. \quad (7)$$

Here, $\hat{\mathbf{Q}}, \hat{\mathbf{K}}$ and $\hat{\mathbf{V}}$ denote the projections of the feature maps $\hat{\mathbf{F}} \in \mathbb{R}^{\hat{C} \times \hat{H} \times \hat{W}}$, which are adjusted to 1/2 or 1/4 the resolution of the original input image to correspond to different scale levels within the U-shaped architecture. These projections are then reshaped to a dimension of $h \times (\hat{C}/h) \times (\hat{H}\hat{W})$, where h denotes the number of attention heads. The paralleled attention heads make CCAT

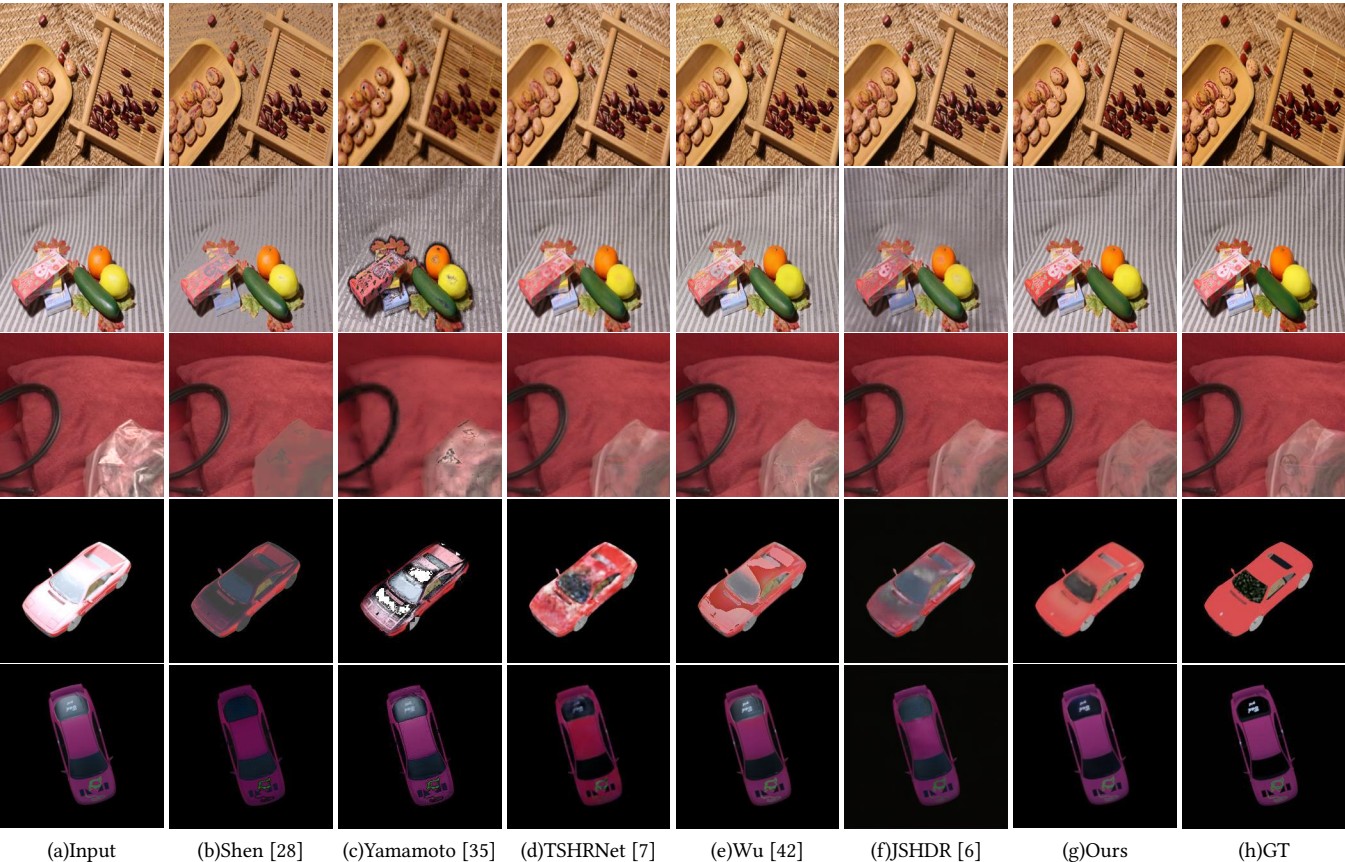

|(a)Input|(b)Shen [28]|(c)Yamamoto [35]|(d)TSHRNet [7]|(e)Wu [42]|(f)JSHDR [6]|(g)Ours|(h)GT|

**Figure 3: Visual comparative analysis of our method against leading SOTA approaches, highlighting our superior ability to remove specular highlights while preserving the original image's color tone, structure, and crucial details, such as text clarity on reflective surfaces.**

capable of capturing different aspects of inter-channel relationships. This diversity allows for a more comprehensive understanding of the feature maps, enhancing the network's ability to discern and emphasize the most relevant features for specular highlight removal.

### 3.4 Adaptive Global Dual Attention Transformer (G-DAT)

Adaptive Global Dual Attention Transformers (G-DAT) are deployed at the bottleneck between the encoder and decoder. Given the higher semantic level at this stage, the information each pixel carries is more abstract and globally contextual than at any other layer in the network. In response to this heightened level of abstraction, we have devised parallel dual attention mechanisms. These are designed to concurrently grasp the intricate inter-channel and inter-pixel dependencies on a global scale, as depicted in the following equation:

$$\text{G-DAT}(\hat{\mathbf{F}}) = \beta \times \text{CCAT}(\hat{\mathbf{F}}) + (1 - \beta) \times \text{PSAT}(\hat{\mathbf{F}}). \quad (8)$$

Here, the considerable reduction in the size of each feature map $\hat{\mathbf{F}}$ due to multiple downsampling stages implies that even the Pixel-wise Self-Attention Transformer (PSAT), which directs attention on a global scale, does not impose a significant computational load.

The operational methodology of PSAT is akin to that detailed in Equation (3); however, unlike the attention in Equation (3) which is confined within a specific window, PSAT expands its focus to cover a global range. Additionally, since the input to the bottleneck module lacks certain visual details, we do not incorporate spectral domain elements in PSAT. This approach ensures that the computational demand is directly proportional to the spatial dimensions of each feature map, maintaining linear computational efficiency.

### 3.5 Objective Function

To optimize the performance of our specular highlight removal methodology, our goal is to align the output as closely as possible with the ground truth diffuse images. This entails not only achieving pixel-level accuracy but also preserving key image attributes such as luminance, contrast, and structure. To accomplish this, we introduce a composite objective function that integrates both Mean Squared Error Loss ($\mathcal{L}_f$), which accounts for pixel-level fidelity, and Structural Similarity Loss ($\mathcal{L}_s$), focusing on maintaining structural integrity. The formulation of our overall objective function is as follows:

$$\mathcal{L} = w_1 \cdot \mathcal{L}_f + w_2 \cdot \mathcal{L}_s, \quad (9)$$

where $w_1$ and $w_2$ have been empirically set to 1 and 0.4, respectively.

We incorporate the Structural Similarity Index Measure (SSIM), as proposed by Wang et al. [46], into the structural similarity component $\mathcal{L}_s$ of our objective function, as delineated below:

$$\mathcal{L}_s = 1 - \frac{(2\mu_D\mu_G + C_1)(2\sigma_{DG} + C_2)}{(\mu_D^2 + \mu_G^2 + C_1)(\sigma_D^2 + \sigma_G^2 + C_2)}, \tag{10}$$

where $\mu_D$ and $\mu_G$ represent the average pixel values of the output diffuse image D and the target ground truth image G, $\sigma_D^2$ and $\sigma_G^2$ are the variances of images D and G, ad $\sigma_{DG}$ is the covariance between D and G. $C_1$ and $C_2$ are constants to stabilize the division with weak denominators.

## 4 Experiments

### 4.1 Benchmark

In the pursuit of advancing the real-world applicability of Specular Highlight Removal (SHR) networks, the utilization of real-world datasets is crucial. To date, the PSD [5] dataset stands out as the only comprehensive collection across various objects where both highlight samples and their corresponding diffuse ground truths are captured in real-world scenarios. However, PSD's limitation lies in its repetition of scenes across different polarization angles, reducing sample diversity despite having nearly 10000 pairs. To overcome the limitations of dataset size and enhance the robustness of deep learning methods for SHR, we propose the creation of a hybrid benchmark. This benchmark amalgamates real-world samples with synthetically generated samples that adhere to optical principles, offering a balanced and comprehensive training and testing environment.

Our hybrid benchmark encompasses data from three distinct datasets: PSD [5], SHIQ [6], and SSHR [7], each serving a different purpose. The PSD dataset provides real-world photographs of both specular and diffuse samples, making it a valuable resource for realistic training. Contrastingly, SHIQ's real-world specular samples are paired with diffuse images created via the RPCA method [47], once state-of-the-art. Despite possibly no longer being the leading technique, its use enriches training diversity, highlighting its ongoing relevance. Meanwhile, the SSHR dataset offers a fully synthetic collection, created with open-source rendering software, adding a valuable dimension of diversity. For training, we selected 9481 pairs from PSD, 9825 pairs from SHIQ, and a random subset of 10000 pairs from SSHR. For testing, we used the official dataset splits for the PSD and SHIQ datasets (947 pairs for PSD and 1,000 pairs for SHIQ). However, due to the large size of the SSHR test split (18,000 samples), a random subset of 1,000 samples was selected to ensure dataset size parity with PSD and SHIQ. This selection strategy guarantees a diverse and balanced collection of samples for both the training and testing phases, ensuring robustness.

To our knowledge, this is the first instance where a hybrid benchmark combining multiple datasets has been employed for SHR. This approach not only enriches the training and testing environment but also sets a new standard for future research in the field, potentially enhancing the performance and generalizability of SHR methods across a wider range of real-world and synthetic scenarios.

Table 1: The quantitative comparison results, arranging traditional methods in the upper section and learning-based approaches below. The highest-performing results are emphasized in bold, while the second-best are underscored.

| Method | PSD (947images) | | | SHIQ (1000images) | | | SSHR (1000images) | | |
|---|---|---|---|---|---|---|---|---|---|
| | PSNR↑ | SSIM↑ | LPIPS↓ | PSNR↑ | SSIM↑ | LPIPS↓ | PSNR↑ | SSIM↑ | LPIPS↓ |
| Tan [26] | 5.44 | 0.218 | 0.746 | 5.47 | 0.483 | 0.823 | 10.87 | 0.778 | 0.357 |
| Yoon [48] | 16.09 | 0.498 | 0.325 | 19.34 | 0.679 | 0.471 | 28.47 | 0.916 | 0.094 |
| Shen [27] | 19.56 | 0.666 | 0.238 | 24.77 | 0.890 | 0.200 | 24.53 | 0.896 | 0.101 |
| Shen [28] | 21.33 | 0.753 | 0.142 | 27.30 | 0.917 | 0.102 | 24.00 | 0.891 | 0.094 |
| Yang [29] | 4.74 | 0.250 | 0.893 | 5.31 | 0.556 | 0.837 | 10.72 | 0.781 | 0.358 |
| Shen [30] | 11.51 | 0.324 | 0.360 | 12.24 | 0.491 | 0.473 | 27.13 | 0.914 | 0.077 |
| Akashi [31] | 17.48 | 0.565 | 0.334 | 21.78 | 0.700 | 0.460 | 29.46 | 0.924 | 0.076 |
| Huo [49] | 20.16 | 0.767 | 0.182 | 23.80 | 0.909 | 0.154 | 18.62 | 0.804 | 0.281 |
| Fu [34] | 15.24 | 0.688 | 0.146 | 16.40 | 0.724 | 0.306 | 26.15 | 0.910 | 0.076 |
| Yamamoto [35] | 18.37 | 0.541 | 0.274 | 25.49 | 0.858 | 0.201 | 26.95 | 0.902 | 0.094 |
| Saha [36] | 15.98 | 0.455 | 0.314 | 22.05 | 0.832 | 0.287 | 23.38 | 0.886 | 0.110 |
| SLRR [39] | 13.25 | 0.571 | 0.235 | 14.74 | 0.724 | 0.283 | 26.16 | 0.916 | 0.060 |
| JSHDR [6] | 22.78 | 0.811 | 0.089 | **37.97** | **0.980** | **0.034** | 26.43 | 0.301 | 0.059 |
| SpecularityNet [5] | 23.58 | 0.838 | 0.085 | 30.92 | 0.963 | 0.058 | 31.07 | 0.941 | 0.041 |
| MG-CycleGAN [43] | 22.12 | 0.815 | 0.085 | 26.80 | 0.935 | 0.091 | 28.40 | 0.874 | 0.092 |
| Wu [42] | 23.93 | 0.863 | 0.062 | 31.57 | 0.965 | 0.059 | 33.45 | 0.951 | 0.028 |
| TSHRNet [7] | 23.30 | 0.826 | 0.097 | 34.57 | 0.972 | 0.044 | 33.32 | 0.950 | 0.036 |
| AHA [45] | 20.79 | 0.845 | 0.084 | 21.42 | 0.903 | 0.165 | 31.57 | 0.944 | 0.035 |
| Ours | **25.28** | **0.883** | **0.049** | 33.81 | 0.975 | 0.039 | **36.48** | **0.964** | **0.023** |

* JSHDR's source code is not publicly available; the results are obtained from an executable file provided by its authors.

### 4.2 Evaluation Metrics

In our study, we utilize a suite of full-reference evaluation metrics to assess performance, including Peak Signal-to-Noise Ratio (PSNR), Structural Similarity Index (SSIM) [46], and Learned Perceptual Image Patch Similarity (LPIPS) [50]. For PSNR and SSIM, higher scores denote better performance, indicating a greater similarity between the generated images and the ground truth. On the other hand, a lower score in LPIPS suggests enhanced visual quality, as this metric measures the perceptual similarity between generated and reference images in a way that is more aligned with human visual perception.

### 4.3 Implementation Details

Our model is implemented using PyTorch and trained using the Adam optimizer with default parameters on NVIDIA H800 GPU. We employ standard settings for the Adam optimization algorithm, utilizing a batch size of 8 and training for 100 epochs, starting with an initial learning rate of $2e-4$ and progressively diminishing it to $1e-6$. To enhance the robustness and generalizability of our model, we incorporate a comprehensive set of data augmentation techniques. These augmentations include random cropping of images, resizing, horizontal and vertical flipping, and the application of the mixup strategy to generate composite images from the original data, thereby exposing the model to a diverse range of variations and improving its ability to handle different data effectively.

### 4.4 Comparisons with State-of-the-Art Methods

To conduct a thorough evaluation of our Specular Highlight Removal (SHR) method relative to the current state-of-the-art, we compared our model against a total of 18 representative SHR techniques, which comprise both 11 traditional and 7 learning-based approaches. For the traditional methods, we processed the test samples directly to obtain their output results. To guarantee a fair comparison, we retrained all the learning-based models on the same benchmark dataset compiled for our study. During this retraining

process, we adhered to the training settings (loss, iterations, hyper-parameters, etc.) as specified in the original publications of each method.

*4.4.1 Quantitative Comparison.* Table 1 showcases the quantitative performance of various specular highlight removal methods across three datasets, utilizing three distinct evaluation metrics. Our model, DHAN-SHR, demonstrates superior performance overall, with the sole exception being the results of JSHDR [6] on the SHIQ dataset, which was released in conjunction with JSHDR. It's important to note, however, that JSHDR's source code is not publicly available; our analysis is based on results obtained from an executable file provided by its authors. This limitation means we couldn't retrain JSHDR under the same conditions as other methods, diminishing the comparative value of its performance on the SHIQ dataset. Notably, JSHDR shows significantly lower performance on both the PSD and SSHR datasets compared to DHAN-SHR, with the gap being particularly pronounced outside the SHIQ dataset.

The performance of our DHAN-SHR model on the PSD and SSHR datasets outpaces other methods by a considerable margin, especially highlighting its exceptional performance on the PSD dataset. This dataset, known for its real-world, high-resolution images, underlines the adaptability and effectiveness of our model in real-world application scenarios, suggesting DHAN-SHR's robust capability in addressing specular highlight removal across diverse conditions.

*4.4.2 Qualitative Comparison.* The visual comparisons between our DHAN-SHR and SOTA methods, which include the top 2 traditional and top 3 deep learning methods based on average metric data in Table 1, are illustrated in Figures 3. For optimal clarity, it is recommended to zoom in.

Observations from Figure 3 reveal that our method excels not only in removing specular highlights effectively—surpassing even the reference ground truth in the third row—but also in preserving the original tone and consistent color of the entire image. Remarkably, it maintains the detail in diffuse areas and restores clarity to details previously obscured by reflections. In contrast, the methods we compared often fail to fully eliminate highlights, sometimes resulting in black spots within the treated areas. More problematic are the visual effects noted in the fourth row, where these methods disrupt the image's original structure and details, leading to poor visual outcomes. Furthermore, in the fifth row, while competing methods tend to erase or blur text on the car's rear window, our approach successfully retains and sharpens these details.

## 4.5 Ablation Studies

To assess the efficacy of our model's integral features, ablation studies were executed, utilizing the averaged metrics from the PSD, SHIQ, and SSHR test sets to ensure a robust evaluation. To better understand the cumulative effect of adding modules and their interaction, we constructed the model starting from a backbone UNet with the same down-sample depth, overall architecture, and number of modules as our full model. Importantly, the total parameter count of the starting backbone UNet is 4.635M, which is more than our full model's 4.533M. This ensures that the effectiveness of our method is not solely due to an increase in parameter count. When

**Table 2: Ablation results.**

| U-Net | P_SSSWAT | C_SSSWAT | FP | CCAT | PSAT | PSNR↑ | SSIM↑ | LPIPS↓ |
|---|---|---|---|---|---|---|---|---|
| ✔ | ✗ | ✗ | ✗ | ✗ | ✗ | 29.79 | 0.930 | 0.048 |
| ✔ | ✗ | ✗ | ✗ | ✔ | ✗ | 30.95 | 0.933 | 0.043 |
| ✔ | ✗ | ✗ | ✗ | ✔ | ✔ | 31.48 | 0.935 | 0.041 |
| ✔ | ✔ | ✗ | ✗ | ✗ | ✗ | 30.30 | 0.932 | 0.044 |
| ✔ | ✗ | ✔ | ✗ | ✗ | ✗ | 30.36 | 0.931 | 0.045 |
| ✔ | ✔ | ✔ | ✗ | ✗ | ✗ | 30.54 | 0.931 | 0.044 |
| ✔ | ✔ | ✔ | ✔ | ✗ | ✗ | 30.77 | 0.933 | 0.044 |
| ✔ | ✔ | ✔ | ✔ | ✔ | ✔ | **31.86** | **0.940** | **0.037** |

adding modules to the backbone, we simply replaced the original UNet module with our module at the corresponding position.

The summarized findings, detailed in Table 2, underscore our full model's superiority. Specifically, P_SWAT and C_SWAT correspond to Pixel-wise and Channel-wise Shifting Window Attention Transformer, respectively. When we add the Frequency Processor (FP) to them, they evolve into the Pixel-wise Spatial-Spectral Shifting Window Attention Transformer (P_SSSWAT) and Channel-wise Spatial-Spectral Shifting Window Attention Transformer (C_SSSWAT), respectively. CCAT (Channel-Wise Contextual Attention Transformer) is the basic component in the downsampled encoding and decoding layers. Additionally, we combine CCAT and PSAT (Pixel-wise Self-Attention Transformer) in the bottleneck to create the Adaptive Global Dual Attention Transformer (G-DAT).

From Table 2, we observe that CCATs, which appear most frequently, contribute the most improvements (3.89% PSNR) compared to the backbone UNet. Cooperating PSAT with CCAT in the bottleneck captures an additional 1.78% PSNR improvement, indicating that pixel-level and channel-level global attention can compensate for each other and achieve significant progress. When using P_SWAT and C_SWAT separately in the top level of the U-shape, we obtain 1.71% and 1.91% improvements, respectively, demonstrating their individual effectiveness. Their combined 2.52% improvement further highlights their combinational strength. Importantly, incorporating the Frequency Processor (FP) into the pure spatial domain architecture results in a 3.29% improvement, emphasizing the benefits of hybrid attention that combines spectral and spatial information. Our full model successfully integrates all these individual components, achieving superior metric results. The ablation study affirms the importance of combining these elements for optimal specular highlight removal performance.

## 5 Conclusion

In this study, we introduce the Dual-Hybrid Attention Network for Specular Highlight Removal (DHAN-SHR), a novel approach that effectively addresses the challenge of specular highlight removal in multimedia applications. DHAN-SHR leverages novel adaptive hybrid attention mechanisms, excelling at capturing both local and global dependencies, and at the same time, incorporating spectral domain features to effectively model complex interactions between specular highlights and surface properties. We assembled an extensive benchmark dataset combining images from three different highlight removal datasets. Experimental results demonstrate that DHAN-SHR outperforms 18 state-of-the-art methods across various test datasets, both quantitatively and qualitatively.

## Acknowledgments

This work was supported by the Science and Technology Development Fund, Macau SAR (Grants 0141/2023/RIA2 and 0193/2023/RIA3), the National Key Research and Development Program of China (No. 2023YFC2506902), the National Natural Science Foundations of China (Grant 62172403), and the Distinguished Young Scholars Fund of Guangdong (Grant 2021B1515020019).

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
