# OpenReview forum: "Dual-Hybrid Attention Network for Specular Highlight Removal"
_acmmm.org/ACMMM/2024/Conference — MM2024 Poster_

### Official Review · Reviewer_z9A4 · 2024-05-20

**Rating:** 3
**Confidence:** 3

**Summary:**

This paper introduces the Dual-Hybrid Attention Network for Specular Highlight Removal (DHAN-SHR), featuring novel attention mechanisms that capture spatial and spectral information, thereby enabling the accurate removal of specular highlights and restoration of diffuse components.  In addition, a new benchmark dataset has been compiled by combining images from three datasets, resulting in 29,306 training pairs and 2,947 testing pairs.  The study retrains and evaluates 18 state-of-the-art methods on this dataset and claim that DHAN-SHR outperforms existing methods both quantitatively and qualitatively.

**Strengths:**

This paper designs two new attention mechanisms and develops an end-to-end network inspired by U-Net. Additionally, it combines three datasets into a large benchmark and conducts extensive experiments to validate the effectiveness of their method.

**Limitations:**

1. **Lack of Novelty in Model Design:**

   The proposed Adaptive Local Hybrid-Domain Dual Attention Transformer appears to be a minor modification of the Swin-Transformer. Although the modules look complex, they essentially lack significant innovation. The novelty of the modifications is questionable, as they do not offer substantial advancements beyond existing models. The Adaptive Global Dual Attention mechanism also seems to be a simple alteration of the self-attention mechanism, which is quite common in the computer vision (CV) domain. The changes introduced do not present a breakthrough or novel approach to the existing self-attention frameworks.

2. **Benchmarking and Experimental Validation:**

   The authors combined three datasets (PSD, SHIQ, SSHR) to create a large benchmark for their experiments. While the results show that the proposed method outperforms 18 other methods, the authors need to demonstrate whether the model achieves state-of-the-art (SOTA) performance on the official splits of these three datasets. Results on the combined benchmark alone are insufficient to validate the method's general applicability and superiority, as the data split might lead to unfair evaluations.

3. **Ablation Study and Analysis:**

   In the ablation study, the authors only present the impact of removing single modules on the results. There is no analysis on the combined removal of multiple modules, which could provide deeper insights into the contributions of each component. Moreover, the authors merely state the results of the ablation experiments without thoroughly analyzing the reasons behind the performance changes when specific modules are removed.

4. **Complexity in Model Representation:**

   The model structure diagram is overly complex, with each module named in full like Pixel-wise Spatial-Spectral Shifting Window Attention and Channel-wise Spatial-Spectral Shifting Window Attention, and the text containing numerous abbreviations like SSSWA, CCAT. This makes it difficult for readers to follow and understand the model architecture. Simplifying the diagram and maintaining consistency in the use of abbreviations would enhance readability.

5. **Confusion in Notation:**

   There is confusion regarding the notation used in Equation 9, specifically whether the alpha and beta parameters are the same as those in Equations 1 and 8. This lack of clarity in notation can lead to misunderstandings and should be addressed to ensure the mathematical consistency of the paper.

**Suitability:**

2

---

### Official Review · Reviewer_NHaZ · 2024-05-27

**Rating:** 5
**Confidence:** 3

**Summary:**

The paper proposes a method for Specular Highlight Removal. The network composes of several modules including Adaptive Local Hybrid-Domain Dual Attention Transformer which then consists of Pixel-wise Spatial-Spectral Shifting Window Attention Transformer.

The paper also proposes to do the attention in the Frequency domain for better results.

The paper then evaluates the proposes architecture on a hybrid dataset consisting of existing datasets of both real-world and synthetic images. The architecture ranks top quantitatively except on one dataset where it ranks the second.

**Strengths:**

1. The paper is well-written, with the figure of the network architecture very illustrative.
2. The proposed architecture is pretty novel and the result validates the effectiveness of the architecture.
3. The ablation studies give evidence that the introduced modules are essential in the model's performance.

**Limitations:**

The writing can be clearer in terms of the equations, for example in equation (1) and (8) alpha and beta appear to be a learnable weighting parameter, which as fas as I understand is different from the balancing factor in the loss (Equation 9). Consider changing this to avoid confusion.

**Suitability:**

3

---

### Official Review · Reviewer_eXcM · 2024-05-28

**Rating:** 5
**Confidence:** 2

**Summary:**

This paper proposes the Dual-Hybrid Attention Network for Specular Highlight Removal (DHAN-SHR), which removes specular highlights from images without relying on additional priors or supervision. DHAN-SHR uses Adaptive Local Hybrid-Domain Dual Attention Transformers (L-HD-DAT) and Adaptive Global Dual Attention Transformers (G-DAT) to capture and process information across different scales and domains, achieving efficient highlight removal. A large-scale benchmark dataset was compiled, and experiments demonstrated that DHAN-SHR outperforms the state-of-the-art methods across multiple test datasets and evaluation metrics, significantly improving image quality and consistency.

**Strengths:**

- The proposed L-HD-DAT captures local features effectively in high-resolution feature maps through parallel pixel-level and channel-level self-attention mechanisms.

- DHAN-SHR achieves efficient specular highlight removal without relying on highlight masks or other prior information, enhancing the method's practicality and generalization ability.

- By transforming features between the frequency and spatial domains, it enhances the model's ability to handle the details and complex reflective areas.

- The method was extensively evaluated on multiple comprehensive datasets and demonstrated better performance against the state-of-the-art methods.

**Limitations:**

- The introduction of multiple attention mechanisms and feature processing modules may result in longer training times. I am curious about the speed of training and inference of the proposed approach compared to the baseline—also the computational complexity of each component.

- What's the difference between the CCATs within the encoder and decoder pathways, as the authors used different colors?

- How are the input embedding and the contextual optimizer in Fig. 1 implemented?

**Suitability:**

2

---

### Meta-Review · Area_Chair_3jKH · 2024-07-01

**Recommendation:** Accept (Poster)
**Confidence:** 5

**Metareview:**

The paper initially got (2) weak accepts and (1) borderline reject. The main concerns are about the novelty of the model design and the complexity of the model representation. However, after rebuttal, all concerns have been addressed.

Based on their feedback, the decision was made to recommend it for acceptance to ACMMM 2024. We congratulate the authors on their acceptance!